# Occupational Hearing Loss for Platinum Miners in South Africa: A Case Study of Data Sharing Practices and Ethical Challenges in the Mining Industry

**DOI:** 10.3390/ijerph19010001

**Published:** 2021-12-21

**Authors:** Liepollo Ntlhakana, Gill Nelson, Katijah Khoza-Shangase, Elton Dorkin

**Affiliations:** 1Faculty of Health Sciences, School of Public Health, University of the Witwatersrand, Johannesburg 2000, South Africa; gill.nelson@wits.ac.za; 2Department of Speech Pathology and Audiology, Faculty of Humanities, School of Human and Community Development, University of the Witwatersrand, Johannesburg 2050, South Africa; Katijah.khoza-shangase@wits.ac.za; 3Anglo American, Johannesburg 2091, South Africa; Elton.dorkin@angloamerican.com

**Keywords:** electronic data, occupational exposures, personal data, audiometry, healthcare providers, machine learning systems, ethical principles

## Abstract

Background: The relevant legislation ensures confidentiality and has paved the way for data handling and sharing. However, the industry remains uncertain regarding big data handling and sharing practices for improved healthcare delivery and medical research. Methods: A semi-qualitative cross-sectional study was used which entailed analysing miners’ personal health records from 2014 to 2018. Data were accessed from the audiometry medical surveillance database (n = 480), the hearing screening database (n = 24,321), and the occupational hygiene database (n = 15,769). Ethical principles were applied to demonstrate big data protection and sharing. Results: Some audiometry screening and occupational hygiene records were incomplete and/or inaccurate (N = 4675). The database containing medical disease and treatment records could not be accessed. Ethical challenges included a lack of clarity regarding permission rights when sharing big data, and no policy governing the divulgence of miners’ personal and medical records for research. Conclusion: This case study illustrates how research can be effectively, although not maliciously, obstructed by the strict protection of employee medical data. Clearly communicated company policies should be developed for the sharing of workers’ records in the mining industry to improve HCPs.

## 1. Introduction

Hearing conservation programs (HCPs) represent an occupational noise induced hearing loss (ONIHL) prevention measure. A company’s HCPs contains workers’ personal information, audiometry medical surveillance, and occupational noise exposure records to monitor risks for ONIHL, the number of workers diagnosed with ONIHL, and the effectiveness of surveillance programs [1]. Globally, HCP data accessed from various occupational settings have been used to highlight the high prevalence of ONIHL as a result of exposure to high noise levels [2]. In a systematic review, Chen et al. (2020) [3] concluded that HCPs records show complex interactions of risks for ONIHL. Nevertheless, prevention measures implemented by various industries have reduced ONIHL morbidity but have not prevented ONIHL [4].

Occupational noise induced hearing loss is a widely researched public health problem [2]. In 2005, the global prevalence of hearing loss due to exposure to noise in the workplace was estimated to range from 7% in high-income countries to 21% in low- and middle-income countries (LMICs) [5] In LMICs, mining, agriculture and construction industries have been reported to produce the most excessive noise exposure levels and to present with the highest prevalence rates of ONIHL [3,5,6]. In South Africa, a LMIC, the mining industry reported a higher prevalence rate of ONIHL than this (30%) in 2016/2017 [7]. Despite noise reduction and other measures (e.g., noise surveillance and audiometry medical surveillance) implemented in South African mines over the last 12 years, the industry continues to struggle to prevent ONIHL [8].

Mines’ HCPs, designed to prevent ONIHL, constitute big data, which focus on noise exposure measurements, as well as noise monitoring and reduction strategies. HCPs also incorporate the premise that noise exposure for any worker should not exceed 85 decibels (dB) averaged over an eight-hour working period (8-h time-weighted average (TWA)) [9]. However, for a HCP to be efficient and effective, other HCP pillars, in addition to noise reduction, must be adhered to [10]. These include administrative controls, HCP training and education, use of hearing protection devices, risk-based medical examinations, and audiometry and medical surveillance [10]. Big data comprise massive volumes of highly diverse information, which includes personal information, and occupational and medical history of individuals and large cohorts [11]. Thus, South African mines’ HCPs datasets constitute copious amounts of miners’ electronic records collected for risk assessment related to occupational exposures and the prevention of associated medical conditions. Opportunities for medical research, using machine learning systems (MLSs) and algorithms to predict health risks, may seem limitless for the mining industry, but such opportunities have not been extensively explored [12,13].

Electronic data recording uses MLSs with artificial intelligence (AI) capabilities, which automatically classify and predict outcomes by applying various algorithms to datasets [14]. Since the South African mines use electronic data to monitor occupational exposures and diseases, as well as to support medical decisions, it is imperative that occupational health and safety (OHS) practitioners understand MLSs and the relevant algorithms used for data collection, access, and analysis, to compile OHS compliance reports [3].

The lack of familiarity by end-users of MLSs impacts negatively on the quality of data. The fact that end-users cannot measure outcomes based on the collected data hinders hearing healthcare delivery, medical decision-making, and HCP planning [3]. Good quality audiometry data and audiometrists who are skilled in interpreting automated audiometry test results are essential. However, the quality of both has been questioned by some researchers in South Africa [15,16]. The quality of audiometry records is a critical indicator of the effectiveness of an HCP. Thus, it is imperative that audiometrists record data meticulously in order to enable accurate analysis for the reporting of valid results to inform medical decisions and to improve hearing healthcare delivery. Thus, consistent and accurate audiometry medical surveillance may not exist in some instances, which negatively affects the quality of HCP data and hinders the accurate prediction and prevention of ONIHL. No studies in South Africa have assessed audiometrists’ MLSs skills or standards of data recording, which are essential for programs aimed at predicting and preventing ONIHL.

High quality data are also essential for robust scientific research regarding HCPs and ONIHL prevention. However, there is no clarity on practices or policies in the mining industry for seamless and transparent data sharing with embedded ethical principles of confidentiality and beneficence. The implementation of the protection of personal information (PoPI) Act of 2013, enacted in 2020 [17], has paved the way for companies in South Africa to maintain the integrity and confidentiality of workers’ records and has set the stage for a national debate on the use of big data protection and sharing for healthcare and research.

Silo reporting (referring to the reluctance to share information with others from different divisions) [18] of risks associated with ONIHL may impede efforts directed towards the prediction and prevention of ONIHL. There are factors, other than noise, that are associated with hearing loss [8,19], such as ageing, being male and black, genetic predisposition, exposure to recreational noise [19], and treatments for tuberculosis and HIV [19,20,21,22,23]. Exposure to chemicals (nitrogen sulphide and carbon monoxide) emitted during some mining processes have also been reported to increase the risk of hearing loss, but research on this is limited in the South African mining industry [24]. Studies using miners’ HCP records have shown that data on common medical conditions (tuberculosis, cancer and HIV) and treatments thereof [22], as well as data on occupational exposures (dust and chemicals) that are associated with ONIHL, are often kept separately from HCP data [16,19].

Data recording and data sharing practices by the mines’ healthcare practitioners are governed by the Health Professions Council of South Africa’s (HPCSA) ethical principles for good practice [25,26] (specifically, confidentiality and beneficence) and the OHS Act (Act No.85 of 1993) [27], respectively. The South African mines, in their annual occupational diseases’ reports submitted to the Department of Mineral Resources and Energy (DMRE) [8], showed a decrease in the number of ONIHL cases from 2015/16. In the same report [8], the South African mines showed a lack of integrated reporting of all occupational exposures (e.g., dust, nitrogen, carbon-monoxide) that are associated with hearing health risks, and interventions to mitigate these risks. The decrease of ONIHL cases in 2015/16 may have been due to reasons other than improvements in HCPs (e.g., practitioners’ lack of training in and understanding of MLSs, which in turn may lead to incomplete HCP records and/or inconsistencies in HCP data capturing) [15,17]. A lack of integrated reporting of HCP findings inhibits the practical implementation of the HPCSA guidelines which guide the recording of patient data (Booklet 9) [25]. There are no evidence-based studies on interactions between ethical rules that govern miners’ health record keeping and the reporting of occupational diseases.

Legally, access to workers’ medical records, e.g., medical conditions and treatments, and personal information, is restricted to medical practitioners employed by the company [26]. Legislation and rules that govern the reporting of, and access to, workers’ data include the PoPI Act of 2013 [17], the National Health Act (Act No.61 of 2003) [28], and Booklet 5 of the HPCSA (Confidentiality: Protecting and Providing Information) [26]. In addition, according to the Health Professions Act, 1974 (Act No. 56 of 1974) [29], healthcare practitioners hold rights and privileges with regard to patients’ personal and medical information. Although individual companies may design specific data access and protection guidelines suitable for their needs and their multidisciplinary context, there may be governance challenges around privacy, transparency, security, and quality assurance of the big data collected by the mines, which should be investigated.

Table 1 summarises the basic ethical principles that guide data sharing for healthcare practitioners, as applied to mine employees’ data.

While the protection of miners’ data is ensured by these acts and rules [17,24,26,28], the mining companies impose additional restrictions to data access for external medical practitioners and researchers, citing the protection of miners’ medical information as the reason for doing so.

In this case study, we reviewed the personal health data (exposure and audiometry data) in an HCP, to investigate the personal data protection processes followed by a South African platinum mine and to describe the use of big data in a MLS, which may lead to ethical pitfalls.

## 2. Materials and Methods

This was a case study of a large platinum mine in South Africa. We reviewed individual miners’ personal and exposure data from electronic audiometry medical surveillance, hearing screening and occupational hygiene records (stored in three datasets) from 2014 to 2018. A semi-qualitative cross-sectional study design was used [31].

The data collection process comprised two phases. First, data were extracted from the three Microsoft Excel databases, which were designed by the mine’s hearing conservation practitioners. Second, the ethical principles applied by the company with regard to data sharing were explored.

Hearing screening (dataset 1, N = 24,321): A hearing screening, to assess risk for hearing loss, is conducted annually on each individual miner by a qualified audiometrist [8] The dataset comprised the bilateral audiogram records of 24,321 miners, and included the results of testing at frequencies of 0.5, 1, 2, 3, 4, 6, and 8 kHz that were used to calculate percentage loss of hearing (PLH) scores and bilateral standard threshold shifts (STSs). Six data points per audiogram (three frequencies per ear) were used to estimate risk for hearing loss (STS).

Audiometry medical surveillance (dataset 2, N = 480): This data set, accessed from the occupational medical practitioner (OMP), contained results from screening audiograms of miners with a PLH > 2.5% from baseline and those of some miners who presented with ear-related conditions (indicating risk for ONIHL) [8]. The dataset comprised 480 miners who were referred to the OMP by the audiometrist, but 10 audiometry medical surveillance records were missing from the OMP’s dataset. Seventy-six miners in the dataset were compensated for ONIHL.

Occupational hygiene (dataset 3, N = 15,769): Occupational hygienists collect data on exposures (including noise) that are associated with adverse health effects [32]. These data were stored in a separate database, which included data on occupations (job titles), noise exposure levels (dBA), and platinum mine dust (PMD) exposure levels (mg/m³). Approximately 96 data points were used to measure noise exposure for individual miners, based on a noise reading every five minutes for an 8-h shift.

Figure 1 illustrates the three datasets used in the study, all of which included the miners’ employee number, age, and sex. These three variables were used to merge the three datasets, using STATA (version 15.1). The numbers in Figure 1 refer to data collected in 2014.

The second phase of data collection comprised completing a checklist, as shown in Table 2, to record information about the ethical principles that were applied to miners’ data reporting and access, guided by Booklet 9 of the HPCSA [25], the National Health Act [28], and the PoPI Act [17]. Information was collected during discussions with the mine’s three hearing conservation practitioners, viz. the OMP, the audiometrist, and the occupational hygienist.

A fourth set of data (dataset 4), which was stored in a QMed format, could not be accessed. QMed is the mine’s integrated big data health management system, which allows for complex machine learning analysis. It is used by the hearing conservation practitioners and contains stricter data access restrictions than those that applied to the three MS Excel datasets. The database comprised a comprehensive record of each miner, which medical practitioners could access, and included key items, such as scheduling for medical examinations, medical surveillance, primary healthcare, anti-retroviral treatment (ART), and disease management data. The QMed system was developed in line with best-practice guidelines with the intention of providing an integrated healthcare management solution [33]. QMed enables: (1) the efficient medical care of miners, (2) medico-legal compliance, and (3) healthcare compliance needs of the mine, in line with the Mines Health and Safety Act [34].

Figure 2 illustrates the ethical considerations followed in the data collection process to show rigor during data collection.

Ethical clearance was obtained from the University of the Witwatersrand’s Human Research Ethics Committee (clearance certificate no. M180273).

## 3. Results

The mine’s hearing conservation practitioners used a two-step process of HCP record keeping. Step 1 involved daily recording of miners’ complete HCP data in MS Excel spreadsheets. In Step 2, some of the miners’ records were transferred to the QMed system. The hearing conservation practitioners maintained both the MS Excel and QMed datasets.

### 3.1. Access to Audiometry Medical and Occupational Hygiene Surveillance Data

Many of the audiometry medical (dataset 1 and 2) and occupational hygiene (dataset 3) surveillance records accessed from MS Excel were incomplete and/or inconsistent. A total of n = 4675 individual miners’ records were excluded from the analysis due to employee numbers which were missing or incomplete. The initial date of employment was missing for some. There were several employees for whom more than one baseline PLH score was recorded without explanation. Moreover, some had a PLH shift score of >50% with no specified follow-up by the OMP (n = 53).

Although permission to use the QMed dataset was initially granted by the mine, access to the data was later denied, based on conditions of confidentiality as stated in the PoPI Act. Information about some of the miners’ ear-related conditions and treatments which would have been accessed from QMed were excluded from the analysis. Miners’ records stored in the QMed system (all miners), which included data on medical conditions such as tuberculosis, HIV, diabetes, cancer, and treatments thereof, could not be accessed due to restrictions applied to external users for research purposes.

### 3.2. Accessing Miners’ Audiometry Medical Surveillance Records—Ethical Challenges

Table 2 illustrates data reporting, accessing, and the application of the ethical principles of justice, beneficence, and confidentiality that ensure the protection of miners’ personal and medical information (dataset 1–3). However, this resulted in challenges with regard to accessing data from the MS Excel and QMed databases.

#### 3.2.1. Justice

Fair and equitable treatment for all (miners) was ensured. All workers who were exposed to occupational noise were at risk of developing ONIHL. The miners’ audiometry, medical (dataset 1 and 2), and occupational hygiene (dataset 3) surveillance records were accessed.

#### 3.2.2. Beneficence

The use of MLSs ensured that miners’ audiometry surveillance records (datasets 1 and 2) could be accessed, and complied with the HPCSA guidelines in Booklet 5, Section 9.1.1.1 [26]. The MLSs were used as registries to monitor hearing and ear-related medical conditions, and treatments thereof.

#### 3.2.3. Confidentiality

The confidentiality rule is central to trust between practitioners and patients. Confidentiality of miners’ audiometry medical records was ensured by hearing conservation practitioners by, e.g., password encryption, and all data were anonymised before we received them for analysing.

## 4. Discussion

In this case study, we reviewed the use of personal health data (exposure and audiometry) in an HCP to establish risk assessment practices and explored the application of personal data protection processes used by a South African platinum mine. Our intent was to start a debate on the use of big data in MLSs and data sharing, which may lead to ethical challenges.

Worldwide, occupational hearing health research is impeded by data access restrictions, due to both legislation and guidelines that are designed to ensure the privacy and confidentiality of workers’ information and/or company-specific information. This case study illustrates how research can be effectively, although not maliciously, obstructed due to the over-zealous protection of employee medical data by a company. Specifically, access to (and therefore analysis of) the data of several thousand miners for research aimed at predicting occupational hearing loss was hindered by tensions between the application of ethical principles that govern workers’ medical data, such as ethical guidelines for good practice, confidentiality and the protection of personal information, and access to data for research purposes.

The use of machine learning systems to improve data quality and protect personal information has been legislated in South Africa [17,24], but data automation intended to guide medical decision still presents challenges. The participating mine used MS Excel and QMed to record miners’ data. According to Luy [35], MS Excel is inappropriate for the processing and analysis of large datasets as it lacks programming functions and falls short of the requirements for machine learning. On the other hand, the QMed system allows for big data risk assessment analysis and medical research [36]. Thus, using different MLSs with different capabilities impeded data access and restricted hearing health data analysis required in research. While this was a case study of a single large mine in South Africa, the findings are likely to apply to other mines and, perhaps, industries in this and other countries.

### 4.1. Justice

All miners undergo medical surveillance as a mandatory procedure to identify early signs of occupational diseases. The mines follow risk management protocols that are designed to efficiently monitor all miners and prevent occupational diseases [7,37]. Indeed, the mine in this case study conducted annual audiometry surveillance to identify miners at risk of developing ONIHL, and therefore to prevent ONIHL.

The South African mines are guided by acts (Mine Health and Safety Act No. 29 of 1996; PoPI Act of 2013) to ensure the aspect of justice, where miners have frequent and regular annual medical examinations to reduce occupation-related health risks and, in turn, benefit from best practice medical surveillance [34]. However, the use of different MLSs (rather than a single MLS), viz. Excel and QMed, in this case study, to capture annual audiometry medical surveillance records may hinder the accurate tracking of the miners’ hearing function, subsequent medical decisions, and research efforts related to hearing loss prevention.

Ethical challenge: Big data regarding hearing health medical conditions and treatment for tuberculosis and HIV from the QMed system could not be accessed, which prevented comprehensive data analysis for this case study and possibly restricted the mine’s implementation of actions for the early prediction and prevention of occupational hearing loss.

We recommend that this, and other mines store data in a single database, using one health management system to track the health of individual miners.

### 4.2. Beneficence

The mine HCP reports refer to audiometry surveillance big data records. The HCP reporting is based on occupational noise exposure levels, and hearing deterioration is calculated using a PLH score [38]. Other risk factors, such as tuberculosis and HIV treatments, and ear diseases such as otitis media, otitis externa, and impacted cerumen, were included in the QMed dataset (to which access was refused), but not in the records that were accessed for this study. Miners’ PLH scores can be influenced by these ear- and hearing-related risk factors (tuberculosis, HIV, otitis media, otitis externa, impacted cerumen, and their treatments) [22,23], which may not have been taken into consideration when assessing hearing deterioration and planning ONIHL prevention strategies. While the exclusion of miners’ medical information from the audiometry surveillance records may be based on maintaining the confidentiality of medical records and adhering to the mine’s privacy policies, it hinders efforts to prevent ONIHL [16,19,23].

The Mine Health and Safety Act (MHSA) has legislated the reporting of HCP data to the DMRE to ensure the accurate tracking of miners at risk of ONIHL, and the reporting of cases of ONIHL [34]. However, evidence-based research has shown that the South African mines use different HCP databases, which presents challenges such as missing records and the inconsistent recording of noise measurements and PLH scores [15,17]. The individual mines and the government departments that publish the annual occupational and medical reports should undertake quality assurance of the HCP reports submitted to the DMRE.

Ethical challenges: Incomplete records and inaccurately entered records lead to poor quality surveillance data, which in this case may lead to inequitable hearing healthcare for individual miners. This also has the potential to negatively affect the accurate prediction and prevention of occupational hearing loss.

It is recommended that companies develop and communicate transparent policies regarding big data access and sharing processes with all hearing conservation practitioners and miners, to empower miners to take ownership of their hearing health and to ensure equitable hearing health care and accurate monitoring of hearing health trends.

### 4.3. Confidentiality

The mine adhered to the rules that govern patient confidentially and protection of personal information with regard to occupational medical record keeping [24]. The National Health Act [26], Booklet 5 of the HPCSA [24], the MHSA [34], and the POPI Act [16] clearly state that healthcare professionals are mandated to record health data for all users (patients and/or employees) and to design electronic information systems to ensure the security of health records. By using MLSs for big data storage, the mine complied with these requirements.

The OMP is responsible for ensuring that the mine’s medical surveillance programs and MLSs are designed to monitor miners’ occupational hazards associated with health conditions, to evaluate intervention strategies used to mitigate risk, and to prevent occupational diseases [26]. Furthermore, the MHSA [34], the PoPI Act, and HPCSA [26] all require the application of privacy control measures for the protection of workers’ health records. For example, the mines are mandated to report occupational hazards and medical conditions affecting miners to the DMRE’s medical inspectors, and to ensure that miners’ identifiable information is anonymized [34]. In our assessment of the mine’s HCP records, there was a lack of clarity (in the mines’ annual health and safety reports) about which database(s) was used to report audiometry medical surveillance results to the DMRE, although confidentiality and protection of miners’ records were maintained [15,16]. The PoPI Act had not been effected when our study was conducted, but the mine’s medical officers applied the PoPI Act restrictions when denying our request to access the miners’ medical records for research purposes.

Ethical challenge: Criteria used in the decision to share miners’ confidential personal and medical records for medical research are provided in Booklet 5 of the HPCSA [26]. The application of the confidentiality rule is at the discretion of the mine’s medical officer (Section 3.2), but this was not clear to the OMP in this case study. Restricted access to records for miners’ medical conditions and treatments thereof may have been due to a misinterpretation /misunderstanding of Section 3.2 of Booklet 5 [26].

We recommend a transparent application of the privacy and confidentiality rules to all hearing conservation practitioners and employees who handle miners’ data, to encourage big data sharing. Moreover, we recommend that health care practitioners are knowledgeable about the legislation pertaining to confidentiality and their understanding thereof.

### 4.4. General Observations

There are multiple ethical principles (Figure 3) for the protection of personal and medical information used by the mine’s hearing conservation practitioners for record keeping, reporting of ONIHL cases, and accessing data for research regarding ONIHL prediction and prevention. The PoPI Act [17] ‘carries more weight’ than the National Health Act and the HPCSA ethical guidelines [25,26,29]. Thus, the application of these ethical principles and data protection regulations shows respect for individual worker’s rights with regard to the protection of their medical information, which was ensured by the healthcare professionals (OMP and audiologist).

Miners’ audiometry hearing results are reported in such a way that they are easy to understand. For example, results are relayed as ‘pass’ (hearing within normal limits) or ‘refer’ (hearing problem, which may require further intervention, e.g., refer to OMP or refer for a repeat hearing screening). However, research has indicated that miners have limited understanding and knowledge regarding their hearing function [39]. Thus, although the hearing screening results may be easily understood by the miners, the use of four separate datasets by hearing conservation practitioners may hinder the understanding of the negative effects of other diseases and ear-related conditions (and their treatments) on hearing function, by both the miners and the health practitioners. Miners should be encouraged to understand their hearing health status from the audiometry medical surveillance programs. To facilitate this, audiometry results should be integrated with medical records and presented to miners in an easy-to-understand manner.

### 4.5. Limitations

This case study was based on findings from one large-scale mine in South Africa, and the challenges experienced might not apply when conducting research at other mines, large or small. However, it is likely that other mines have the same or similar approaches to data sharing for research purposes. Reported in our previous studies, big data collection and sharing using different MLSs in this case brought about missing data, which in turn restricted data analysis and the accurate prediction of ONIHL for these miners.

### 4.6. Recommendations

Mining company employees should be well-informed about, and comprehensively understand, the legislation, rules, guidelines, and policies that govern sharing of employee data for research (and other) purposes. While data sharing should be guided by ethical rules and company policies, the implementation of these rules and policies should ensure efficient surveillance programs and encourage medical research.

All databases that contain miners’ health and safety records should be complete, accurate, comprehensive, and integrated to ensure efficient risk management frameworks for occupational hazards, which will benefit all miners. The mines should consider implementing quality checks to ensure high quality data.

Mining companies should review their policies about data sharing for research purposes and communicate those policies clearly to employees. Such policies, stipulating access rights for researchers outside of the company, will encourage scientific research.

## 5. Conclusions

Efficient MLSs that are used to store and manage HCP big data depend on good quality data, accurate and consistent data input, clear checklists for all ethical principles, and comprehensive company regulations related to the recording of, and access to, data for medical research.

In this case study of one mine, there were discrepancies in the implementation of HCP checklists used for record keeping, which rendered big data inefficient for the prediction and prevention of ONIHL. In addition, there were ethical challenges that arose due to unclear data sharing practices, which, in turn, potentially hindered medical decisions regarding ONIHL prevention. There is a need for all the ethical principles to be integrated in the recording and sharing of big data, together with HCP checklists, in order to assist in the prediction and prevention of occupational hearing loss. This case study illustrates how research can be effectively, although not maliciously, obstructed due to the over-zealous protection of employee medical data by a company. The findings may apply to other mines and industries in South Africa and other countries. Debates around the mines’ HCP big data should incorporate privacy and confidentiality, as well as the right to data access and sharing in order to improve medical research, which is aimed at preventive healthcare.

## Figures and Tables

**Figure 1 ijerph-19-00001-f001:**
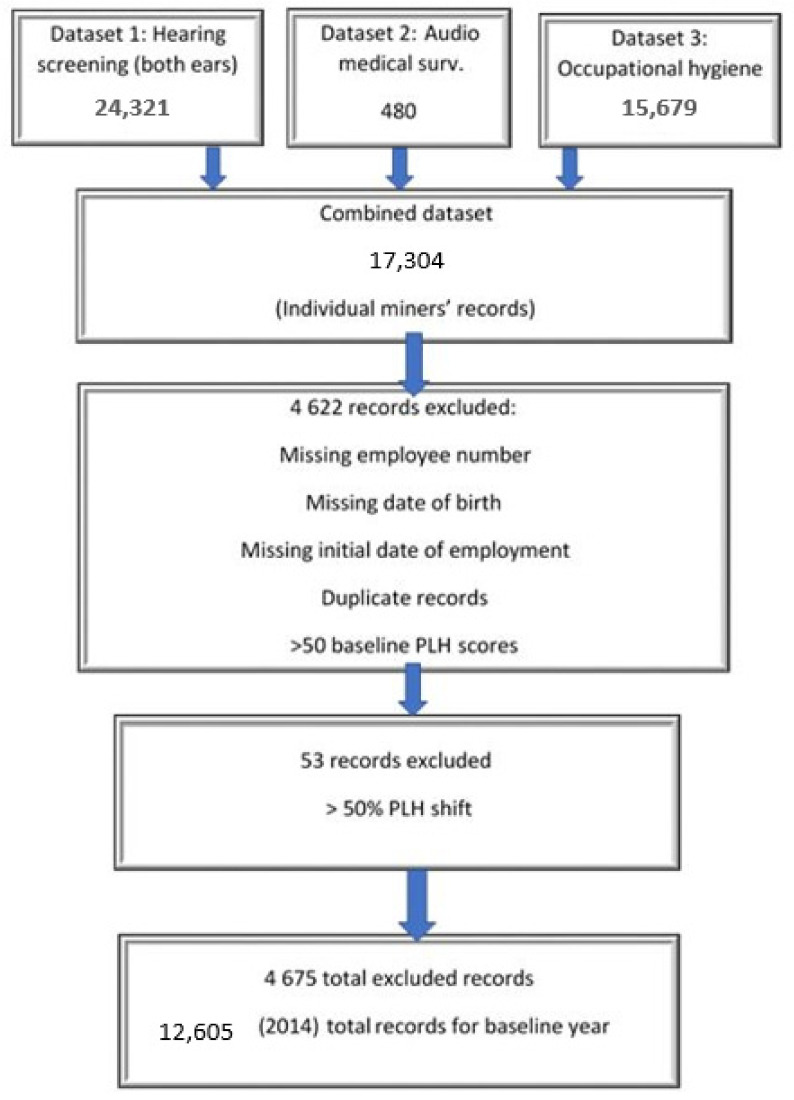
Data management process followed showing individual miners’ records included and those excluded from merged dataset.

**Figure 2 ijerph-19-00001-f002:**
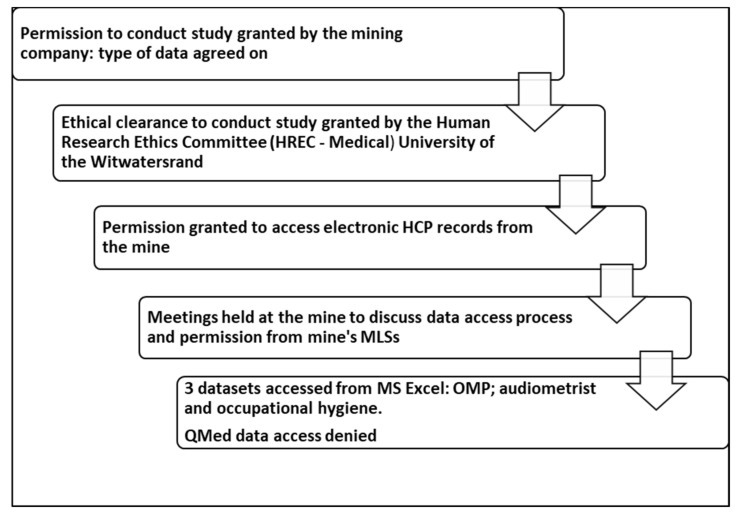
Ethical considerations in the data collection process (Human research ethics committee (HREC); Hearing conservation programme (HCP); Machine learning system (MLS); Occupational medical practitioner (OMP)).

**Figure 3 ijerph-19-00001-f003:**
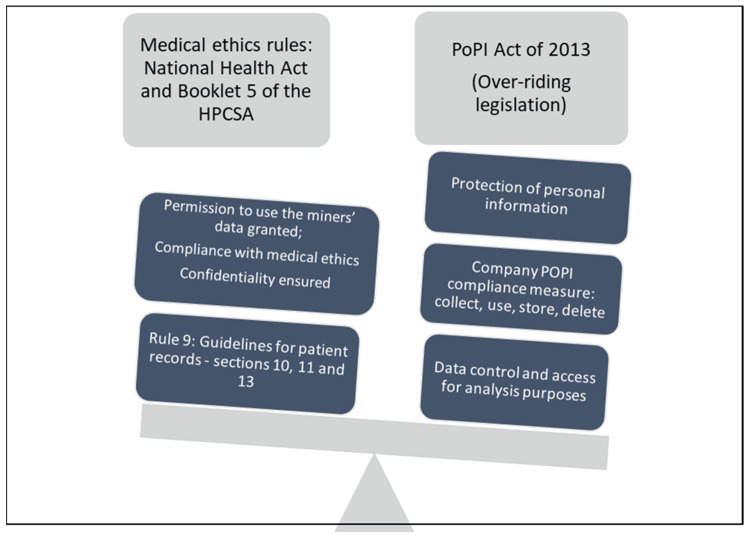
Illustration of medical ethics and personal protection regulations for access to personal data in MLSs (Protection of personal information (POPI).

**Table 1 ijerph-19-00001-t001:** Summary of ethical principles.

Ethical Principle	Definition	Explanation
Confidentiality	Personal information will be held in confidence. Guidelines in Booklet 5 of the HPCSA state that a practitioner may divulge patient information under certain confidential conditions.	Miners’ personal and medical information must be confidential and sharing of the medical information may be at the discretion of the medical practitioner.
Beneficence	Duty to do more good than harm	Research using miners’ audiometry data must be shared confidentially and miners’ identity must be protected to address scientific questions.

Adapted from Flite and Harman (2013) [30].

**Table 2 ijerph-19-00001-t002:** Checklist of ethical considerations used from data accessed.

		Dataset to which Ethical Considerations Applied		
Ethical Consideration	Guiding Document	1	2	3
Confidentiality	*†‡	√	√	√
Anonymity	*	√	√	√
Protection of personal information	‡	√	√	√
Complete and consistent records	*†‡	√	√	√
Standardized electronic records	‡	√	√	√
Computer stored records	*†‡	√	√	√
Data access for research (external users)	*†‡	√	√	√

* HPCSA; † National Health Act; ‡ PoPI Act.

## Data Availability

Data sharing not applicable. No new data were created and analyzed in this study. Data sharing is not applicable to this article.

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
