# Peer review of "Occupational Hearing Loss for Platinum Miners in South Africa: A Case Study of Data Sharing Practices and Ethical Challenges in the Mining Industry"

_ijerph, 2021, doi:10.3390/ijerph19010001_

Round 1
Reviewer 1 Report
The authors don't have partially addressed my comments. See the below.
- Again, “Hearing conservation programs" is defined at the beginning of the sentence, so, use the acronym as follows;
p.2 L55-56: correct "Hearing conservation programs also incorporate the premise..." to "HCPs also incorporate the premise..."
- Line 60: the definition is provided…hearing protection devices (HPDs).
Delete "(HPDs)." Avoid acronym that is never used in the subsequent text.
- Again, make sure to provide the definition of acronyms used in Figure 2, as the footnote. Spelling out an acronym in a figure/table is useful for those readers who wish to scan the figures/tables before deciding whether to read the full paper.
- Line 251: Delete references [25,30].
Author Response
Dear Editor,
Thank you for providing us with reviewers' comments. Please see our points in responding to comments from Reviewer 1. Please see the manuscripts with tracks attached for your attention.
-
Again, “Hearing conservation programs" is defined at the beginning of the sentence, so, use the acronym as follows;
Response: Line 52 - 53: has been edited to..."HCP also incorporates..."
p.2 L55-56: correct "Hearing conservation programs also incorporate the premise..." to "HCPs also incorporate the premise..." -
- Line 60: the definition is provided…hearing protection devices (HPDs).
Response: Line 58: HPDs has been deleted.
Delete "(HPDs)." Avoid acronym that is never used in the subsequent text. -
Again, make sure to provide the definition of acronyms used in Figure 2, as the footnote. Spelling out an acronym in a figure/table is useful for those readers who wish to scan the figures/tables before deciding whether to read the full paper.
Response: Line 208: At the bottom of Figure 2, next to the figure title, the footnote has been included. Line: 383 - Figure 3 for POPI we also included a footnote. - Line 251: Delete references [25,30].
Response: reference have been deleted.
Kind regards,
Liepollo Ntlhakana
Reviewer 2 Report
I do not have more comments. Thanks.
Author Response
Dear Editor,
Thank you for sharing Reviewer 2' comments with us.
There were no comments from Reviewer 2 to respond to.
Kind regards,
Liepollo Ntlhakana
Reviewer 3 Report
I have no more comments and remarks
Author Response
Dear Editor,
Thank you for sharing Reviewer 3 comments with us.
There are no comments to respond to from Reviewer 3.
Kind regards,
Liepollo Ntlhakana
This manuscript is a resubmission of an earlier submission. The following is a list of the peer review reports and author responses from that submission.
Round 1
Reviewer 1 Report
This research provides ethical challenges when using the public databases on occupational hearing loss through the case study for platinum miners in South Africa.
This study also gives important insight not only applying for the case of miners but also spreading for general open-science issues. Hence, this paper will have a significant contribution to subsequent relevant research. I have a few comments or suggestions below.
Minor comments
- Results section. In my understanding, Section 3.1. and 3.2 seem to show the results on Dataset 2 and Dataset 3, except Dataset 1. Make sure to clarify the relationship between the three Datasets and the access/ethical issues.
- Don't cite the literature when describing the Results. Make sure to respect the elementary rules of scientific writing.
- I'm a little confused about the association between the Results of 3.2 and Table 2. Table 2 shows that all ethical issues are met in all three databases. Confirm that the Results accurately mirror the Materials and Methods and vice versa. Please clarify the association.
- 7 Line 238: What do you mean "The MLSs were used as registries to monitor hearing and ear-related medical conditions and treatments thereof."? What kind of roles can Machine Learning Systems play for registries? Please clarify it.
- There are too many abbreviations throughout the text. The use of acronyms should be kept to a minimum. As you know, once you defined it, you can use it in the subsequent text. “Hearing conservation programs” shown in P.2 Line 51-52 was already defined in Line 31. "HPDs" shown in Line 57 appears only in there in the text. Don't use an acronym without defining it. For example, To take into account the international and diverse readership, the "PoPI Act (Line 90)" and “HPCSA ” should be explained. Make sure to check all acronyms used in the text and try to minimize the usage.
- Make sure to provide the definition of acronyms used in Figure 2, as the footnote.
- Delete “Please add:” in Line 415.
- Inconsistent style of titles of articles (capitalization rule) and pages, i.e, No.3 uses different capitalization rule from others. No.9 has inconsistencies like "Occupational Hearing loss... " using a small letter for "loss." No.9, and No.12 have no page information provided. Make sure to re-check all the literature information listed in the References. The authors should use a tool such as EndNote or Mendeley for reference management and formatting.
I'm so happy if the above comments would help you when revising your valuable manuscript.
Reviewer 2 Report
I admit the paper is written and organized well. But the title of the manuscript is misleading as I do not see any statistical analysis results shown in the paper. That is, at the first glance of the title, I thought the authors would provide some empirical evidence, eg. predictive modeling, to justify their arguments. In the end, I realized the authors were mainly discussing the data collection and policy, etc. Another big concern is that, authors excluded 4,675 records from the merged data, which might bias the future analysis results. Because the missing data are not necessarily missing at random.
A minor comment: please correct the citation format on Page 1 Line 7 'Chen et al.(2020)[3]'.
Reviewer 3 Report
The work is well-done, however, after reading the whole text I am a little bit disappointed. In my opinion, this text has just practical meaning and no more.
Detailed presentation of methods used by the authors is a strength of this study but the weakness is lack of (examples at least) of analysis of original results, so it is difficult to assess more precious the manuscript. So the title is very promising but the content is disappointing.